# SLIDECHAT: A LARGE VISION-LANGUAGE ASSISTANT FOR WHOLE-SLIDE PATHOLOGY IMAGE UNDERSTANDING

## ABSTRACT

Despite the progress made by multimodal large language models (MLLMs) in computational pathology, they remain limited by a predominant focus on patch-level analysis, missing essential contextual information at the whole-slide level. The lack of large-scale instruction datasets and the gigapixel scale of whole slide images (WSIs) pose significant developmental challenges. In this paper, we present SlideChat, the first vision-language assistant capable of understanding gigapixel whole-slide images, exhibiting excellent multimodal conversational capability and response complex instruction across diverse pathology scenarios. To support its development, we created SlideInstruction, the largest instruction-following dataset for WSIs consisting of 4.2K WSI captions and 176K VQA pairs with multiple categories. Furthermore, we propose SlideBench, a multimodal benchmark that incorporates captioning and VQA tasks to assess SlideChat's capabilities in varied clinical settings such as microscopy, diagnosis. Compared to both general and specialized MLLMs, SlideChat exhibits exceptional capabilities, achieving state-of-the-art performance on 18 of 22 tasks. For example, it achieved an overall accuracy of 81.17% on SlideBench-VQA (TCGA), and 54.15% on SlideBench-VQA (BCNB). We will fully release SlideChat, SlideInstruction and SlideBench as open-source resources to facilitate research and development in computational pathology.

## 1 INTRODUCTION

Computational pathology aims to improve the analysis of digitized tissue samples, such as whole slide images (WSIs), by applying artificial intelligence to aid in the diagnosis, identification, and understanding of disease (Song et al., 2023). Recently, the development of this field has gained rapid momentum, mainly driven by breakthroughs in the visual foundation model (Chen et al., 2024b; Xu et al., 2024a; Vorontsov et al., 2024). These models learn generalized representations by pre-training on large-scale data and perform well in various downstream tasks, including rare cancer detection and biomarker prediction. Building on this base, integration with the powerful Large Language Models (LLMs) further advances the development of the Multimodal Large Language Model (MLLMs) (Lu et al., 2024b), which has made great strides in responding to more complex, open-ended visual queries, enabling it to serve as a versatile assistant at various stages of medical care, including clinical decision-making, education, and research (see Figure 1).

Nevertheless, there are three major challenges that hinder the development and use of pathology MLLMs for real-world clinical applications. First, it is challenging to develop a MLLMs architecture that can effectively capable of gigapixel whole slides (*e.g.*, 100,000 × 100,000 pixels). Existing models (Lu et al., 2024b; Sun et al., 2024; Seyfioglu et al., 2024) often process whole slides by extracting small patch/ROI-level data for subsequent analysis, resulting in limited understanding of global slide context and suboptimal performance in some complex pathological analysis. Second, publicly available multi-modal pathology slide dataset are relatively scarce and of varying quality (Guo et al., 2024; Chen et al., 2023; 2024a), which limits the development of MLLMs trained on such data. Third, current pathology MLLMs (Lu et al., 2024b) are developed using closed-source data from medical institutions. Consequently, the model weights and associated instructional

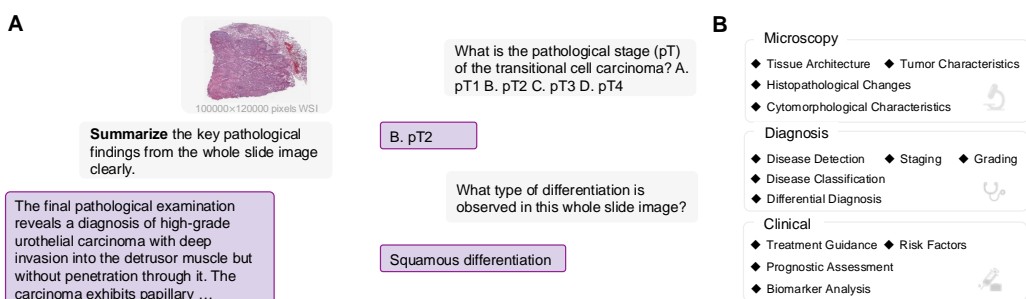

Figure 1: SlideChat' is the first large vision-language assistant specifically designed for whole-slide pathology analysis. SlideChat can generates comprehensive descriptions of whole-slide images and provides contextually relevant responses across various applications.

datasets are typically not made full publicly available, thereby restricting their broader applicability in clinical research and applications.

In this paper, we present SlideChat, the first open-source vision-language assistant capable of understanding gigapixel whole-slide images. First, SlideChat is trained on SlideInstruction, a large-scale multi-modal instruction dataset encompassing data from The Cancer Genome Atlas (TCGA) (Hutter & Zenklusen, 2018) via our specifically designed data processing pipeline (see Figure 3 (A)). SlideInstruction contains 4,181 WSI-caption pairs and 175,754 visual question-answer pairs from 3,294 patients, covering 10 cancer types. The question-answer pairs include both open-ended and closed-ended questions, further divided into 13 subcategories, covering a diverse range of clinical tasks such as tumor grading. SlideInstruction is more than 20 times larger than previous public instruction datasets in the number of instructions(see Table 11 in Appendix). Second, we propose SlideChat, a novel architecture in LLaVA style for capable multi-modal analysis of gigapixel whole slides. As shown in Figure 2, a gigapixel whole slide is first divided into a series of patches, each of which is individually processed by a patch-level encoder to extract local features. The resulting long sequence of feature tokens is then processed by a slide-level encoder employing sparse attention to aggregate the slide-level features. Finally, the aggregated features are fed into a Large Language Model via a projector, which processes user queries and generates responses.

To systematically evaluate the performance of SlideChat in real-world scenarios, we establish a comprehensive digital pathology benchmark (see Table 1) encompassing more than 20 clinical tasks, using data from both TCGA and the in-the-wild Early Breast Cancer Core-Needle Biopsy (BCNB) dataset. This resulted in three test sets: SlideBench-Caption, comprising 734 WSI-caption pairs; SlideBench-VQA (TCGA), comprising 7,827 VQA pairs covering 13 different tasks; and SlideBench-VQA (BCNB), including a total of 7,274 VQA entries from 1,058 patients, covering seven different tasks. Additionally, we compare the performance of SlideChat on another externally proposed dataset, WSI-VQA (Chen et al., 2024a), to further validate its effectiveness. We compare our model with the currently available state-of-the-art general and medical-specialized MLLMs including GPT-4o, LLava-Med (Li et al., 2024a), MedDr (He et al., 2024). Benefiting from large-scale, high-quality training and effective local-global context modelling, SlideChat achieves state-of-the-art performance on 18 out of 22 tasks, with significant improvements over the second-best method 10% on 9 tasks on four benchmarks. Specifically, SlideChat achieves an average accuracy improvement of 13.47% over the second-best model on SlideBench-VQA (TCGA), an average improvement of 12.71% on SlideBench-VQA (BCNB), and an improvement of 5.82% on WSI-VQA. Finally, to accelerate research progress in digital pathology, we make SlideChat fully open-weight, including source code and model weights as well as instruction and benchmark dataset. The key contributions are summarized four-fold in the following:

- We create SlideInstruction, a largest comprehensive WSI instruction-following dataset containing 4.2K WSI-caption pairs and 176K VQA pairs.

- We develop SlideChat, the first vision-language assistant capable of understanding gigapixel whole-slide images, achieving state-of-the-art performance on multiple benchmarks.

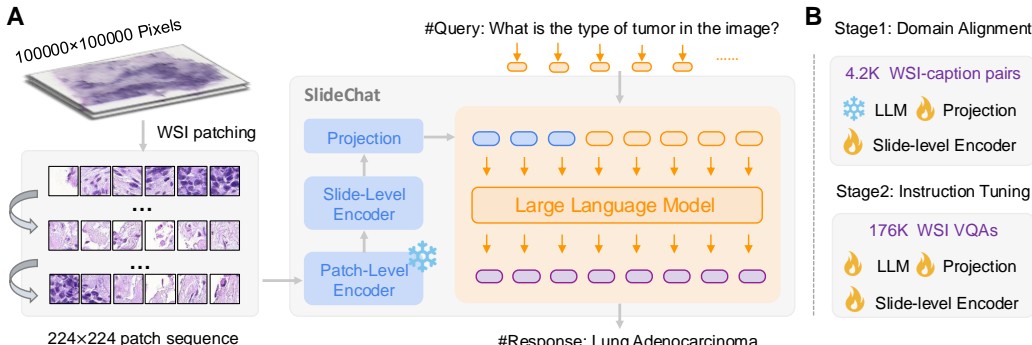

Figure 2: Overview of our SlideChat. (A) SlideChat serializes each input WSI into a sequence of 224×224 patches, converting each into visual embeddings with a patch-level encoder. A slide-level encoder then interacts with these features to generate contextual embeddings. Then, a multimodal projector maps the visual features from the slide-level encoder into a unified space, aligned seamlessly with the LLM. (B) SlideChat was trained for two stages: Cross-Domain Alignment and Visual Instruction Learning.

- We establish SlideBench, a WSIs multi-modal benchmark comprising SlideBench-Caption, SlideBench-VQA (TCGA), and SlideBench-VQA (BCNB), covering 21 different clinical tasks.
- We will release SlideChat, SlideInstruction and SlideBench as open-source resources to facilitate research and development in computational pathology.

## 2 RELATED WORKS

**Whole Slide Image Analysis**    Whole slide images are pivotal in modern pathology, enabling comprehensive analysis of tissue samples for tasks such as predicting patient prognosis, classifying cancer subtypes, and identifying biomarkers (Song et al., 2023; Shao et al., 2023; Li et al., 2024b; Spronck et al., 2023). Recent studies have leveraged pathology foundational models (Wang et al., 2024; Ahmed et al., 2024; Xu et al., 2024b) to enhance WSIs analysis, either through fine-tuning for specific downstream tasks or by employing zero-shot prediction approaches in CLIP (Radford et al., 2021) style. Although these models are effective in task-specific applications, their reliance on fine-tuning or limited zero-shot capabilities restricts their generalizability across diverse and complex user instructions.

**MLLMs in Computational Pathology**    The paradigm of MLLMs enables to effectively respond to more complex, open-ended visual queries while processing pathology image, thus providing significant value across various medical stages. PathChat (Lu et al., 2024b) is a vision-language assistant designed for pathology, developed with 450K private instruction pairs to handle both visual and natural language queries. QuiltInstruct (Seyfioglu et al., 2024) is a large-scale dataset comprising 107K question-answer pairs. Building on QuiltInstruct, Quilt-LLAVA (Seyfioglu et al., 2024) is a model designed for diagnostic reasoning across multiple image patches, leveraging its extensive question-answer pairs to accurately interpret complex H&E data. PathAsst (Sun et al., 2024) combines a pathology-specific CLIP model with Vicuna-13b (Chiang et al., 2023) to create a multimodal generative foundational model tailored for pathology. However, current MLLMs primarily focus on patch or region-of-interest (ROI) data, limiting their utility for slide-level clinical applications where broader contextual understanding is crucial.

## 3 SLIDECHAT

### 3.1 ARCHITECTURE

To achieve the goal of analyzing gigapixel whole-slide images in a multimodal setting, as shown in Figure 2, SlideChat consists of four key designs: the patch-level encoder, the slide-level encoder,

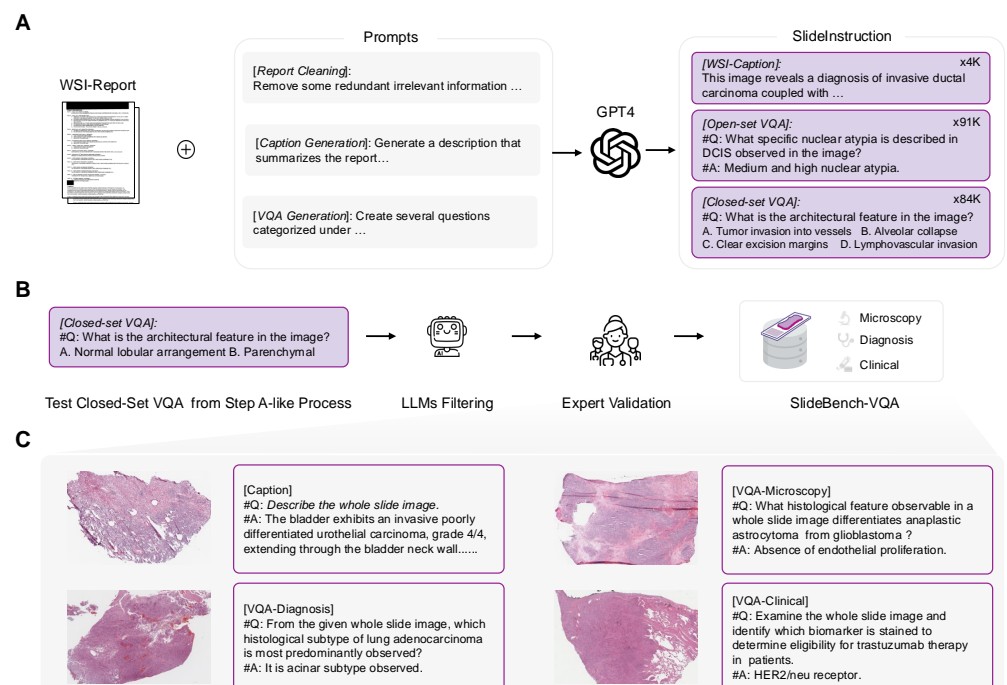

Figure 3: (A) Overview of the SlideInstruction generation pipeline. We prompt GPT-4 to extract the WSI-Caption, Open-set VQA and Closed-set VQA from reports. (B) For the generated Closed-set VQA, we employ LLMs to filter low-quality QA pairs and involve pathologists for validation, resulting in the creation of SlideBench-VQA. (C) Examples of WSI caption and instruction-following scenarios in microscopy, diagnostics, and clinical applications. For additional examples, please refer to Figure 6 in the Appendix.

the multimodal projector module, and the large language model. Our method starts by partitioning the WSI into smaller 224 × 224 pixel patches, making it computationally feasible to process such large images. These patches are then passed through a well-trained, frozen patch-level encoder (Lu et al., 2024a), which extracts localized features from each individual patch, capturing fine-grained details such as cellular structures. Building on this, we employ LongNet (Ding et al., 2023; Xu et al., 2024a) as slide-level encoder to enhance the patch-level embeddings and capture global patterns across the entire slide. This encoder uses sparse attention mechanisms to aggregate both local and global contextual information, enabling the model to perceive intricate local features while capturing the broader context, which is critical for comprehensive pathological assessments. Following the slide-level encoding, SlideChat incorporates a multimodal projection layer that maps these aggregated visual features into a unified space aligned with the LLM. This ensures that the visual features extracted from the WSIs are effectively transformed into representations compatible with the language model, facilitating seamless integration and interaction between visual and textual data. Concurrently, the model accepts natural language instructions from users, such as "What is the type of tumor in the image?". These textual queries are processed by the LLM, which comprehends the textual input and integrates it with the visual features extracted from the WSIs, enabling accurate and contextually relevant diagnostic responses. This multimodal reasoning capability allows SlideChat to provide accurate and contextually relevant answers to complex pathology-related questions, thereby supporting clinical decision-making, education, and research across various medical stages.

## 3.2 DATA

**SlideInstruction** There is a notable lack of large-scale multimodal pathology datasets supporting the training of vision-language assistants for whole-slide image understanding. To support the training of SlideChat, we develop SlideInstruction, a comprehensive instruction dataset, sourced from the

Table 1: Statistical information of SlideBench.

| Subset | #Patient | #Data | #Tumor | #Tasks | Answer Type | Evaluation Metirc |
|---|---|---|---|---|---|---|
| SlideBench-Caption | 734 | 734 | 10 | 1 | Free From | BLEU, Rouge, GPT score |
| SlideBench-VQA (TCGA) | 732 | 7,827 | 10 | 13 | A/B/C/D | Accuracy |
| SlideBench-VQA (BCNB) | 1058 | 7,274 | 1 | 7 | A/B/C/D | Accuracy |

TCGA database, comprising 4,915 whole slide image (WSI)-report pairs from 4,028 patients. Figure 3 illustrates our entire data curation pipeine. We initially prompt GPT-4 to refine the pathology reports, clean up the noise in the report including unrelated symbols, technical details of pathology department procedures, specimen handling and processing information, redundant administrative or legal statements, and some repeated information. For the refined pathology reports, we further employ GPT-4 to generate high-quality multimodal data, comprising two main components: (1) *WSI-Caption Data*: We craft concise, clinically relevant summaries for each whole slide image by prompting the language model to extract key pathological findings. These summaries were structured into coherent paragraphs that highlighted crucial clinical details such as diagnostic results, tumor characteristics, margin status, and lymph node involvement, ensuring the caption dataset is both focused and informative. (2) *WSI Instruction-Following Data*: To enhance the model's ability to follow instructions and improve its comprehension of pathology images, we leveraged GPT-4 to generate tailored question-and-answer pairs for each WSI report. Drawing inspiration by PathChat (Lu et al., 2024b), we structure these questions into three "broad" categories—microscopy, diagnosis, and clinical considerations—which represent key stages in the pathology workflow, and thirteen "narrow" categories focusing on specific aspects within each stage (Figure 1 B). Our carefully crafted prompts are detailed in Appendix A.2.2. To create a comprehensive instructional dataset, we generated two open-ended and two closed-ended QA pairs within each narrow category for every WSI report. Regarding the train/test split, it is worth noting that the WSI-report datasets from TCGA includes two types: (a) one report linked to multiple WSIs, and (b) one report linked to a single WSI. For type (a), where specific diagnostic details may not align perfectly with each WSI, we include all WSIs in the training set to introduce some "noisy data", which can enhance model robustness. For type (b), 80% of WSIs are allocated to the training set and 20% to the test set. Finally, there are 4,181 WSIs for training and 734 WSIs for testing. Consequently, we construct a large-scale training set named SlideInstruction, comprising 4,181 WSI captions and 175,753 instruction-following VQA pairs across various broad and narrow categories.

**SlideBench** To systematically evaluate the performance of SlideChat, We incorporate the remaining 734 WSI captions along with a substantial number of closed-set VQA pairs to establish evaluation benchmark. First, we construct a test set named SlideBench-Caption based on the WSI-Caption data to evaluate the model's ability to generate accurate and coherent descriptions of whole slide images. Secondly, we construct SlideBench-VQA (TCGA) based on closed-set visual question-answering (VQA) pairs along with test WSIs, aiming to evaluate various aspects of model performance. As shown in Figure 3 (B), to improve the quality of the testing benchmarks, we employ four advanced large language models, including GPT-4 (Achiam et al., 2023), InternLM2-Chat-7B (Cai et al., 2024), Qwen-7B-Chat (Bai et al., 2023), and DeepSeek-7B-Chat, to filter closed-set VQAs by predicting answers based solely on the question text. Any questions for which at least three of these models provided correct answers are subsequently excluded. Following this automated filtering, five expert pathologists are invited to review and amend the remaining questions. The review process are guided by the following criteria: (1) Whether the correct answer necessitates image interpretation; (2) Whether the question and its corresponding answer are logically and coherently structured; and (3) Whether the question aligns appropriately with the designated broad and narrow categories. QA pairs failing to meet these criteria are excluded by the pathologists. Consequently, the SlideBench-VQA (TCGA) comprises 7,827 VQAs across 13 categories, with some examples illustrated in Figure 3 C. Additionally, we incorporate the in-the-wild Early Breast Cancer Core-Needle Biopsy (BCNB) WSI dataset (Xu et al., 2021), which encompasses a diverse patient population and a variety of clinical task labels, to enhance the test set benchmark and more comprehensively assess the model's generalization capabilities. In detail, we convert the BCNB dataset into a VQA format by rephrasing the classification objectives into a specific template as questions, while transforming the original multi-class labels into selectable options, and integrate it into SlideBench as an external subset, named SlideBench-VQA (BCNB). This dataset comprises 7,247 VQA pairs

Table 2: Captioning performance across different methods on SlideBench-Caption. Slide (T) refers to the WSI thumbnail with size of $1024 \times 1024$.

| MLLMs | Input | BLEU-1 | BLEU-2 | BLEU-3 | BLEU-4 | Rouge-L | GPT-score |
|-------|-------|--------|--------|--------|--------|---------|-----------|
| GPT-4o | Patch | 0.16 | 0.03 | 0.01 | 0.01 | 0.13 | 1.54 |
| GPT-4o | Slide (T) | 0.10 | 0.03 | 0.01 | 0.01 | 0.11 | 1.03 |
| MI-Gen | Slide | 0.37 | 0.24 | 0.15 | 0.10 | 0.25 | 4.14 |
| SlideChat | Slide | 0.37 | 0.21 | 0.12 | 0.08 | 0.24 | 4.11 |

from 1,058 patients, specifically designed to evaluate SlideChat's zero-shot generalization capability across 7 distinct classification tasks. More detailed information about SlideBench is provided in Table 1.

### 3.3 TWO-STAGE TRAINING

**Stage 1: Cross-Domain Alignment.** SlideChat adopts a two-stage training approach (see Figure 2 B). In the first stage, the primary objective is to align the large language model's (LLM) word embeddings with the visual features extracted from whole slide images. This alignment enables the LLM to interpret visual representations from the slide-level encoder, facilitating the effective utilization of the intricate features within the slides. During this stage, SlideChat is trained to generate descriptive captions using 4.2K WSI-caption pairs from SlideInstruction. Specifically, only the slide-level encoder and projection matrix are updated, while the patch-level encoder and LLM weights remain fixed.

**Stage 2: Visual Instruction Learning.** In the second stage, we focus on visual question-answering tasks to train the model to accurately respond to domain-specific questions concerning whole slide images. During this phase, the model develops the ability to handle a broad range of multimodal instructions, enabling it to generate answers by effectively integrating both visual and textual information. For example, the model must perform various pathology tasks, such as describing the extent of tumor invasion or assessing the degree of cellular differentiation. To accomplish this, we utilize 176K WSI VQAs from SlideInstruction in the second training stage, allowing the slide encoder, projection layer, and large language model components to be fully trainable to ensure comprehensive adaptability. This training approach significantly enhances the model's capability to handle diverse pathology-related tasks, thereby increasing its effectiveness in real-world clinical and research settings.

## 4 EXPERIMENT

We conducted following experiments to evaluate three key aspects of SlideChat: (1) its whole slide image captioning capability, which assesses proficiency in generating descriptive captions that accurately summarize the critical pathological features of a WSI; (2) its visual question-answering (VQA) ability across various complex pathological scenarios and its generalizability in zero-shot settings; and (3) SlideChat's ability to process gigapixel WSIs, capturing both essential global context and intricate details, thereby enhancing its performance compared to patch-level multimodal large language models. For WSI captioning baselines, we benchmark against MI-Gen (Chen et al., 2023), a state-of-the-art method specifically designed for this task. Given that existing MLLMs cannot handle the gigapixel scale of whole slide images, we establish baseline comparisons using two approaches: (1) randomly selecting 30 patches from each WSI and inputting them into MLLMs (e.g., GPT-4 (Achiam et al., 2023), LLaVA-Med (Li et al., 2024a), MedDr (Li et al., 2024a)), followed by a majority voting scheme to generate slide-level predictions; and (2) directly inputting a WSI thumbnail, resized to 1024×1024 pixels, into the MLLMs. For VQA tasks, we further evaluate performance by comparing against random prediction baselines and text-only models, thereby assessing the incremental contribution of visual information. Unless otherwise specified, SlideChat is configured with the patch-level encoder CONCH (Lu et al., 2024a), the slide-level encoder LongNet (Ding et al., 2023), and utilizes the Qwen2.5-7B-Instruct (Yang et al., 2024) as LLM.

Table 3: VQA performance with different methods. Slide (T) refers to the WSI thumbnail with size of $1024 \times 1024$.

| MLLMs | Input | SlideBench-VQA (TCGA) | | | | SlideBench-VQA (BCNB) | WSI-VQA* |
|---|---|---|---|---|---|---|---|
| | | Microscopy | Diagnosis | Clinical | Overall | | |
| Random | Text | 24.44 | 24.91 | 26.44 | 25.02 | 24.40 | 24.14 |
| GPT-4 | | 38.28 | 29.09 | 45.00 | 37.25 | 0 | 18.60 |
| GPT-4o | Patch | 62.89 | 46.69 | 66.77 | 57.91 | 41.43 | 30.41 |
| MedDr | | 73.30 | 57.78 | 74.25 | 67.70 | 33.67 | 54.36 |
| LLaVA-Med | | 47.34 | 32.78 | 47.96 | 42.00 | 30.1 | 26.31 |
| GPT-4o | Slide (T) | 38.28 | 23.10 | 43.42 | 34.07 | 0 | 14.03 |
| MedDr | | 70.48 | 52.47 | 72.80 | 64.25 | 35.48 | 50.95 |
| LLaVA-Med | | 45.82 | 27.58 | 40.84 | 37.39 | 0 | 18.79 |
| SlideChat | Slide | 87.64 (+14.34) | 73.27 (+15.49) | 84.26 (+10.01) | 81.17 (+13.47) | 54.14 (+12.71) | 60.18 (+5.82) |

**SlideBench-Caption** We report BLEU, ROUGE, and GPT scores to evaluate caption generation performance in Table 2. For the GPT score, we use GPT-4 to assess the similarity between the generated captions and the ground truth, providing an overall score on a scale of 1 to 10, with higher scores indicating better performance. When utilizing patch-level inputs, GPT-4o generates individual descriptions for each patch, which are subsequently integrated to create the final slide-level caption. However, this approach yields poor performance, as evidenced by a BLEU-1 score of 0.16 and a GPT-score of 1.54. These results suggest that the patch-based method fails to adequately capture the broader context necessary for accurate WSI captioning. When the WSI thumbnail of size 1024×1024 pixels is used as input to GPT-4o, performance decreases further, with a BLEU-1 score of 0.10 and a GPT-score of 1.03. This suggests that while the thumbnail offers a global view of the slide, it may lack the resolution and detail necessary for generating precise and informative captions. In contrast, MI-Gen, a model specifically designed for WSI captioning, demonstrates significantly superior performance across all metrics, achieving a BLEU-1 score of 0.37 and a GPT score of 4.14. Similarly, SlideChat, shows competitive results with a BLEU-1 score of 0.37 and a GPT score of 4.11. These outcomes highlight SlideChat's ability to effectively integrate both local and global information from the slides and confirm its efficiency in describing whole-slide images, as illustrated by several examples shown in Figure 7 in the Appendix.

**SlideBench-VQA (TCGA)** We further evaluate SlideChat's overall performance on the multiple-choice VQA benchmark. The results on SlideBench-VQA (TCGA), presented in Table 3, compare different methods across three domains: Microscopy, Diagnosis, and Clinical, along with an overall performance score. Random selection achieves an overall score of 25.02% accuracy, serving as a baseline for answer distribution but demonstrating poor performance. While GPT-4, relying solely on text input, outperforms random predictions, it continues to struggle with accurately answering questions. When GPT-4o incorporates patch-level inputs, its performance improves markedly, reaching a score of 57.91% and underscoring the crucial role of detailed visual data. However, using a WSI thumbnail results in a lower score of 34.07%, as the reduced detail restricts its ability to deliver precise answers. MedDr performs well, achieving an overall score of 67.70% with patch-level inputs, though this drops slightly to 64.25% when using the slide thumbnail due to the loss of fine visual details. SlideChat outperforms all other methods, attaining a leading overall accuracy of 81.17%, excelling across all categories and significantly surpassing the competition. Even in more fine-grained pathological scenarios, as depicted in the left portion of Figure 4, SlideChat remains the top-performing model across 13 tasks, particularly in areas such as cytomorphological characteristics, histopathological changes, disease detection, disease classification, and staging and grading, which require the identification of complex pathological visual features. Compared with baselines taking some patches or slide thumbnial as inputs, SlideChat has the capability to analyze a significantly greater number of pathological features with enhanced detail, effectively capturing both localized features and overarching global patterns, allowing SlideChat to provide more accurate and nuanced insights into pathological variations. In Figure 9 of the Appendix, we present comparative examples of different methods, highlighting the superior performance of SlideChat. Additionally, Figure 8 in the Appendix showcases examples of SlideChat's continuous dialogue capabilities, demonstrating its effectiveness in facilitating interactive and comprehensive pathological analysis.

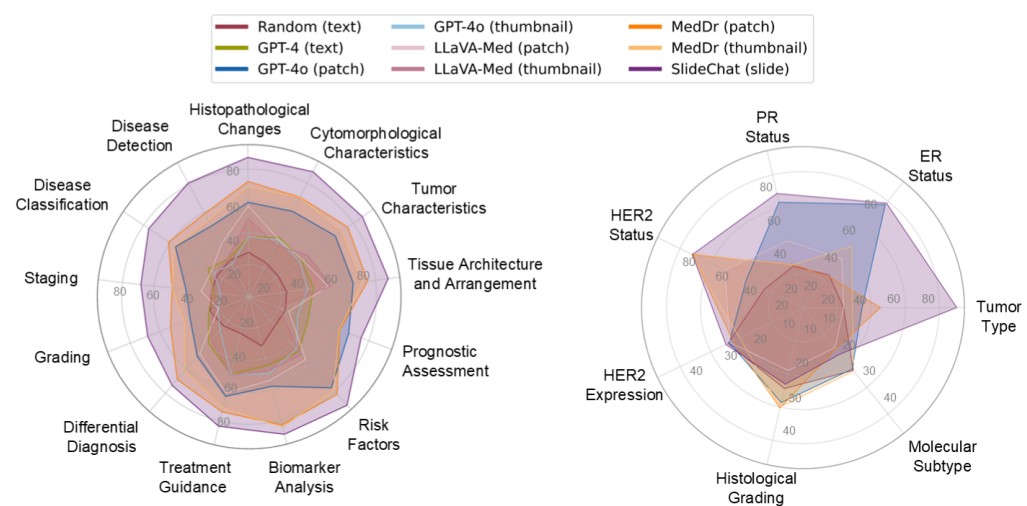

Figure 4: Accuracy on different tasks in SlideBench-VQA (TCGA) (left) and SlideBench-VQA (BCNB) (right).

**SlideBench-VQA (BCNB)** Besides, on SlideBench-VQA (BCNB), we compared SlideChat's zero-shot capabilities with those of different methods. In the zero-shot VQA setting, SlideChat significantly outperforms all other models, achieving the highest overall score of 54.14%. It particularly excels in identifying tumor types, far surpassing other baselines in this task. This performance highlights SlideChat's generalization capability across a wide range of tasks. When taking patches as inputs, GPT-4o outperforms both MedDr and LLaVA-Med, achieving a score of 41.43%, though it still falls short of SlideChat by 12.71%. Notably, GPT-4o and LLaVA-Med performed very poorly, achieving a score of zero across all tasks when evaluated using slide thumbnails from this testing set. MedDr also shows a notable drop in performance when switching from patch to thumbnail inputs, with its overall score falling from 35.48% to 33.67%. This outcome highlights that, for complex WSIs, relying solely on relatively sufficient visual features is inadequate for effectively supporting a range of tasks. Additionally, in the more fine-grained pathological tasks of the BCNB benchmark, as shown in the Figure 4, SlideChat attains state-of-the-art performance on 5 out of 7 tasks, further demonstrates the effectiveness of SlideChat.

**WSI-VQA[*]** We also curated a subset of closed-set VQA pairs from the public WSI-VQA (Chen et al., 2024a) dataset based on our split test WSI list, referred to as WSI-VQA[*], to evaluate the model's performance. SlideChat demonstrates the highest performance with a score of 60.18. Although MedDr performs well with both patch inputs (54.36%) and slide thumbnail inputs (50.95%), it still falls short compared to SlideChat. GPT-4o struggles significantly, especially with slide thumbnails, scoring only 14.03%, which highlights the limitations of using lower-resolution inputs. SlideChat's ability to process whole-slide images allows it to leverage both detailed local information and broader context, making it the most effective model for this benchmark. This further emphasizes its superior capability in handling whole-slide data for VQA tasks.

**Ablation** We performed ablation experiments from different perspectives as follows: a) *Large Language Model Comparison*: We compare the performance of several large language models, each with a parameter scale of approximately 7 billion. The evaluated models include Vicuna-7B-v1.5 (Chiang et al., 2023), Phi-3-Mini-4k-Instruct (Abdin et al., 2024), Qwen1.5-7B-Chat (Bai et al., 2023), Llama3-8B-Instruct (AI@Meta, 2024), InternLM2-Chat-7B (Cai et al., 2024), and Qwen2.5-7B-Instruct (Yang et al., 2024). Specifically, we measured their performance on the SlideBench-Caption task using the GPT-score and their accuracy on three VQA (Visual Question Answering) benchmark datasets. As shown in Table 4, the results demonstrate that SlideChat, powered by the Qwen2.5-7B-Instruct model, achieved the highest performance across all tasks, particularly excelling in the captioning task. These findings underscore the significant potential of developing SlideChat with the Qwen2.5-7B-Instruct model. Utilizing the Qwen2.5 model, we further evaluated

Table 4: Performance Comparison of LLMs and Slide Encoder on WSI Captioning and VQA Tasks.

| LLMs | Slide Encoder | Caption | VQA (TCGA) | VQA (BCNB) | WSI-VQA* |
|---|---|---|---|---|---|
| Vicuna-7B-v1.5 | ✓ | 3.28 | 41.43 | 41.43 | 31.98 |
| Phi-3-Mini-4k-Instruct | ✓ | 2.66 | 79.93 | 43.92 | 60.18 |
| Qwen1.5-7B-Chat | ✓ | 2.92 | 77.63 | 44.07 | 56.89 |
| Llama3-8B-Instruct | ✓ | 3.30 | 78.78 | 42.82 | 55.25 |
| Internlm2-Chat-7B | ✓ | 3.30 | 79.10 | 52.13 | 56.76 |
| Qwen2.5-3B-Instruct | ✓ | 3.40 | 80.32 | 45.79 | 56.38 |
| Qwen2.5-14B-Instruct | ✓ | 3.39 | 82.14 | 51.57 | 60.94 |
| Qwen2.5-7B-Instruct | ✓ | 4.11 | 81.17 | 54.14 | 60.18 |
| Qwen2.5-7B-Instruct | ✗ | 3.95 | 81.21 | 45.49 | 59.67 |

models of varying scales and discovered that larger models generally exhibited superior performance, particularly the 7B and 14B parameter models. While these two models showed comparable performance across the three benchmarks, the 14B model surpassed the 7B model by 2.57% on the SlideBench-VQA (TCGA) task. Given computational resource constraints, SlideChat uses the 7B model by default to achieve the best balance between performance and resource efficiency. b) *Slide-level encoder effectiveness*: We investigate the effectiveness of the slide-level encoder by initially removing it from SlideChat and employing a two-stage training approach. In the first stage, we exclusively trained the projection layers. However, this approach failed to reduce training loss or generate coherent text effectively, likely due to the difficulty of learning the complex visual features of WSIs without the slide-level encoder. With the LLMs frozen and tasked with complex text generation, a simple projection proved insufficient for effectively integrating and aligning visual and textual features. Subsequently, we consider combining data from both stages and training SlideChat without the slide-level encoder by simultaneously updating both the projection layers and the LLM. Under this paradigm, performance on SlideBench-VQA (TCGA) and WSI-VQA (sub), which share the same distribution as the training set, was comparable to the two-stage training configuration with the slide-level encoder. However, a significant decline is observed when evaluating SlideBench-VQA (BCNB), which originates from a different domain; overall performance dropped by over 10% (Table 4), indicating a substantial reduction in the model's generalization ability. Therefore, we recommend incorporating a slide-level encoder to capture the complex visual features of whole slide, as it is particularly effective for cross-domain alignment and enhances the model's generalization performance.

**Interpretability** Despite SlideChat demonstrating promising results, concerns remained regarding the model's perception of large pathological slides. To further assess the model's interpretability, we calculated the correlation between the text output and specific image patches, thereby obtaining patch-level attention scores. By identifying the most significant patches, we gained insights into the precise areas the model focused on during response generation. Highlighting the most relevant regions of the tissue slides not only enhances transparency and bolsters the reliability of the model's outputs but also assists pathologists by directing attention to critical areas requiring closer scrutiny. Ultimately, such interpretability is essential for fostering trust in AI-assisted diagnostics and enhancing the precision and efficiency of clinical evaluations. As shown in Figure 5, we are pleased to observe that the top five important patches identified by the model closely corresponded with the features described in SlideChat's output. Our extraction method retrieves attention weights for patch tokens from each generated token, averaging them across layers and heads. We then identify the top five patch tokens with the highest attention weights for further analysis. For example, in Figure 5 (A), the highlighted patches clearly emphasized regions exhibiting an increased nuclear-to-cytoplasmic ratio, hyperchromatic nuclei, and prominent nucleoli. Similarly, in Figure 5 (B), the selected patches demonstrated areas with dense collagen deposition and reduced cellularity, as detailed in the model's response. This alignment between the highlighted image regions and the textual outputs significantly enhances the model's interpretability, providing increased confidence that it accurately captures and assesses relevant histopathological features. Such consistency deepens our understanding of the model's reasoning processes regarding pathological slides and underscores the potential for integrating these AI systems into clinical workflows with greater assurance.

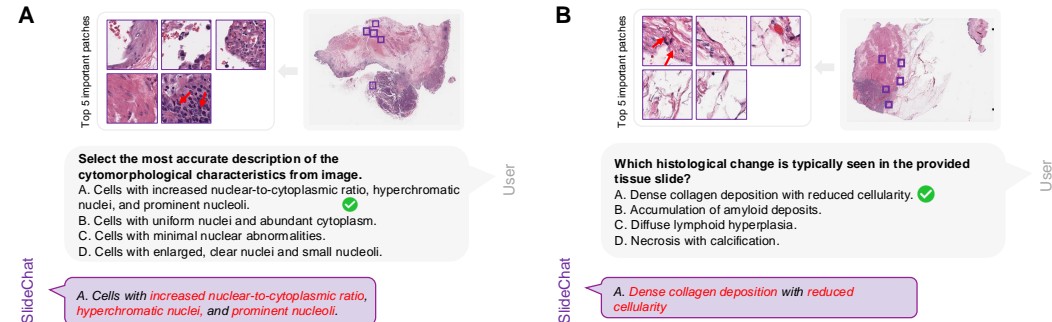

Figure 5: Interpretability and visualization. We identify the top five patch tokens with the highest attention scores associated with the output text responses.

**Computational Cost Analysis**   To evaluate the computational cost of our model architecture, we measured both the inference time and GPU memory consumption throughout the entire pipeline. This pipeline includes the patch-level encoder, slide-level encoder, multimodal projector module, and large language model, all executed on an A100 GPU. After extracting the local and global features of WSIs, the average response time was within 1 second, and GPU memory consumption was approximately 27 GB. The inference time and GPU memory consumption remained well within acceptable limits for gigapixel whole slide images.

## 5   CONCLUSION

In this work, we present SlideChat, the first vision-language assistant capable of understanding gigapixel whole-slide images. Furthermore, we creat SlideInstruction, a largest comprehensive WSI instruction-following dataset to develop SlideChat, as well as SlideBench, a multi-modal benchmark designed to evaluate SlideChat across diverse scenarios. SlideChat demonstrates excellent chat abilities and achieves state-of-the-art performance on 18 tasks.

We bridge the gap between MLLMs and pathology images at the whole-slide level with SlideChat, and believe that it represents a significant advancement towards general pathology and general medical artificial intelligence (GMAI).

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

# Appendix

CONTENTS

# A SLIDEINSTRUCTION AND SLIDEBENCH

## A.1 DATA SOURCE

In this section, we present the sources of the constructed SlideInstruction and SlideBench, which are derived from ten TCGA datasets as well as the BCNB challenge dataset. The Table 5 provides a detailed overview of the specific number of WSIs.

Table 5: Datasets statistics

| Dataset | WSIs | Report | Organ | Purpose |
|---------|------|--------|-------|---------|
| TCGA-BRCA | 1068 | ✓ | Breast | Train, Test |
| TCGA-LGG | 783 | ✓ | Brain | Train, Test |
| TCGA-GBM | 513 | ✓ | Brain | Train, Test |
| TCGA-LUAD | 506 | ✓ | Lung | Train, Test |
| TCGA-LUSC | 474 | ✓ | Lung | Train, Test |
| TCGA-HNSC | 464 | ✓ | Head and Neck | Train, Test |
| TCGA-BLCA | 424 | ✓ | Bladder | Train, Test |
| TCGA-COAD | 419 | ✓ | Colon | Train, Test |
| TCGA-READ | 157 | ✓ | Rectum | Train, Test |
| TCGA-SKCM | 107 | ✓ | Skin | Train, Test |
| BCNC | 1058 | ✗ | Breast | Test |

## A.2 CURATION SCOPE AND PROMPT

In this section, we illustrate the various dimensions of VQAs in SlideInstruction and SlideBench, ensuring comprehensive coverage of diverse pathological scenarios. This includes 3 broad categories and 13 narrow categories. Below are the contents for each category, which help to delineate their scope and meaning, thereby enabling GPT to extract high-quality question-answer pairs more effectively.

### A.2.1 SCOPE

**Microscopy** This category involves assessing the ability to generate microscopy descriptions of pathology images, focusing on clinically relevant features:

- Tissue Architecture and Arrangement: Questions in this category should evaluate the understanding of overall tissue structure and spatial organization within a histological section.
- Cytomorphological Characteristics: These questions should focus on the detailed description of individual cell morphology, including nuclear and cytoplasmic features.
- Tumor Characteristics: Questions under this category should assess the ability to identify and describe features specific to tumors, such as tumor differentiation, invasion, and specific patterns associated with different types of tumors.
- Histopathological Changes: This category should include questions that evaluate the recognition and description of pathological changes in tissue, such as necrosis, inflammation, fibrosis, and other alterations that indicate disease processes.

**Diagnosis** This category tests the ability of models to suggest a reasonable diagnosis based on histological images and relevant clinical context:

- Disease Detection: Questions in this category should evaluate the model's ability to identify the presence or absence of a disease based on histological features and clinical information.
- Disease Classification: These questions should focus on distinguishing between different types or subtypes of diseases, assessing the model's capability to classify conditions accurately based on morphological and histopathological criteria.

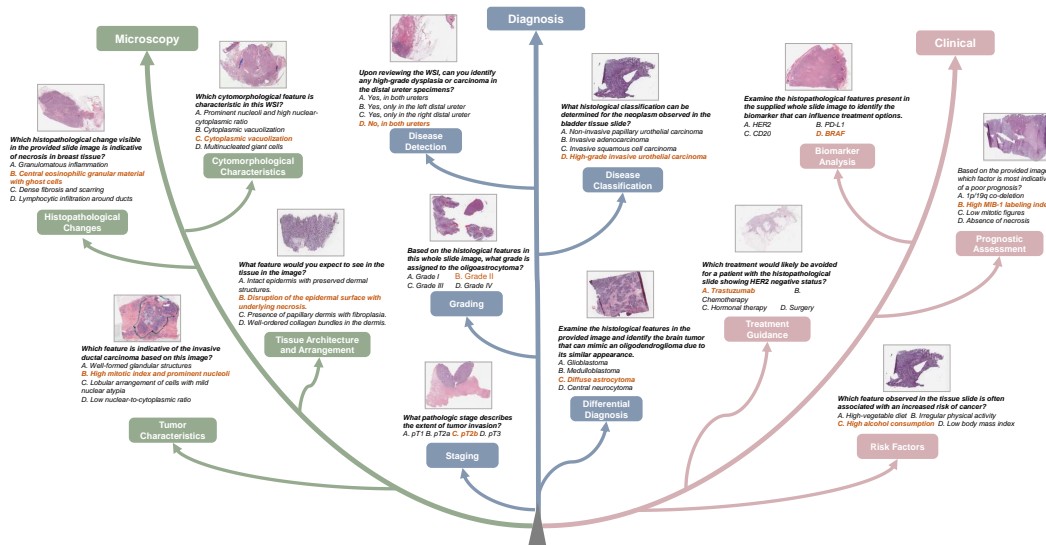

Figure 6: Examples of generated structural VQAs in pathology across Microscopy, Diagnosis, and Clinical scenarios.

- Grading: Questions under this category should assess the model's ability to determine the grade of a disease, particularly tumors, based on the degree of differentiation and cellular atypia observed in histological images.

- Staging: This category should include questions that evaluate the ability to assign a stage to a disease, particularly in oncology, by assessing the extent of disease spread and involvement of surrounding tissues or organs.

- Differential Diagnosis: Questions should test the model's ability to provide a differential diagnosis, distinguishing between multiple potential conditions that may present with similar histological and clinical features.

**Clinical**    This category tests the ability of models to retrieve and apply clinically relevant background knowledge about diseases:

- Treatment Guidance: Questions in this category should assess the model's ability to recommend appropriate treatment options based on the disease in question, considering factors such as disease stage, patient demographics, and any specific clinical guidelines.

- Prognostic Assessment: These questions should focus on evaluating the model's ability to predict the likely course and outcome of a disease, including survival rates, potential complications, and long-term outcomes based on clinical and pathological data.

- Risk Factors: Questions under this category should test the model's knowledge of risk factors associated with specific diseases, including genetic, environmental, and lifestyle factors that may influence disease development or progression.

- Biomarker Analysis: This category should include questions that evaluate the ability to identify and interpret biomarkers relevant to the diagnosis, prognosis, or treatment of diseases, emphasizing their role in personalized medicine and targeted therapy.

### A.2.2    DESIGNED PROMPTS

**Report Cleaning Prompt.**    The prompt used to clean up the report from the original TCGA report is represented in Table 6. This process effectively eliminates extraneous noise from the report, thereby establishing a more solid foundation for caption and QA pairs generation.

Table 6: Prompts for report clean and caption generation.

**[Report Clean Prompt]**    This is the content from the pathology report. Please remove some redundant irrelevant information from the original report, such as technical details of pathology department procedures, Symbols unrelated to the pathological report, specimen handling and processing information, redundant administrative or legal statements, and some repeated information. Show me the cleaned report content.

**[Caption Generation Prompt]**    Based on the above pathological report content, generate a detailed paragraph that summarizes the essential pathological findings. The paragraph should include key information such as the diagnosis, tumor characteristics, margin status, lymph node involvement, and other relevant pathological findings. The summary should not mention the source being a report and should exclude any specific sizes or measurements. The paragraph should be written in a clear and cohesive manner, covering all important points without unnecessary details.

Table 7: Question-Answers generation prompts, including system prompt, general prompt and objective prompt.

**[System Prompt]**    You are an AI assistant proficient in digital pathology. You will receive a pathology report for whole slide images.

**[General Prompt]**    Based on the above pathological report content, your task is to use the provided information, create 2 multi-choice questions amd 2 short-answer questions for each narrow category. The design question should be able to be answered based on the content of the image. Design medical questions very carefully and only ask questions when you are sure of the answer. Answers should be specific and avoid ambiguity. When generating questions, it is necessary to indicate their broad category and narrow category. For multi-choice questions, you should (1) "question type" is "multi-choice questions". (2) Provide the options and answer and reasoning. Provide four answer choices (A, B, C, and D), ensuring that one choice is correct and the other three are plausible but incorrect. (3) Aim to include one answer that is incorrect but very similar to the correct one to increase the difficulty level. For short-answer questions: (1) "question type" is "short-answer questions". (2) Generating questions with different content from multiple-choice questions. For all questions: (1) Do not mention that the information source is report in "question", "anwser". (2) Return JSON format in "question type": xxx, "question": xxx, "options": [], "answer": xxx, "broad category": xxx, "narrow category": xxx for each question. The "options" section is empty for short-answer questions.

**[Objective Prompt]**    Definition of Broad Category and its corresponding Narrow Categories. " The required broad category is Microscopy, which involves assessing the ability to generate microscopy descriptions of pathology images, focusing on clinically relevant features. For the narrow category: Tissue Architecture and Arrangement: Questions should evaluate the understanding of overall tissue structure and spatial organization within a histological section."

Table 8: Prompt for Converting Labels into QA Pairs

**[Label Transformation Prompt]**    Please create prompts for pathology image classification tasks concerning <Task>, transforming traditional labels into a multi-choice question-and-answer format. The original labels include <label 1>, <label 2>, ...

**Caption Generation Prompt.**    The prompt used for caption generation from the refined report is detailed in Table 6, ensuring that the generated caption effectively captures essential summarized information in report.

Table 9: The number of VQA corresponding to each category in SlideBench-VQA (TCGA).

| Broad Category | Narrow Catgory | Number |
|---|---|---|
| Microscopy | Tissue Architecture and Arrangement | 696 |
| | Tumor Characteristics | 562 |
| | Cytomorphological Characteristics | 601 |
| | Histopathological Changes | 633 |
| Diagnosis | Disease Detection | 581 |
| | Disease Classification | 532 |
| | Staging | 671 |
| | Grading | 601 |
| | Differential Diagnosis | 586 |
| Clinical | Treatment Guidance | 597 |
| | Biomarker Analysis | 502 |
| | Risk Factors | 591 |
| | Prognostic Assessment | 674 |

Table 10: The number and options of VQA corresponding to each task in SlideBench-VQA (BCNB).

| Task | Number | Option |
|---|---|---|
| ER Status | 1058 | Postive / Negative |
| HR Status | 1058 | Postive / Negative |
| HER2 Status | 1058 | Postive / Negative |
| HER2 Expression | 1058 | 0 / 1+ / 2+ / 3+ |
| Histological Grading | 926 | 1 / 2 / 3 |
| Molecular Subtype | 1058 | Luminal A / Luminal B / HER2(+) / Triple negative |
| Tumor Type | 1058 | Invasive ductal carcinoma / Invasive lobular carcinoma / Other Type |

**Question-Answers Generation Prompt.** The prompt used to extract QA from reports mainly consist of 4 parts (*i.e.*, <Cleaned Report>+ System Prompt + Objective Prompt + General Prompt), and the detailed content of each part is illustrated in Table 7

**Label Transformation Prompt.** The prompt for transforming BCNB dataset is illustrated in Table 8. We employ GPT to transform individual labels into a question-answer format based on the task type and corresponding classification labels, facilitating the testing of MLLM. For instance, in the context of a tumor type classification task, <Task>represents "Tumor Type", while <label 1>, <label 2>, and <label 3>are "Invasive ductal carcinoma", "Invasive lobular carcinoma", and "Other Type", respectively, enabling the generation of relevant QA pairs.

## A.3 DATA STATISTICS

We have compiled statistics on the number of VQA instances for each category within SlideBench VQA (TCGA) in Table 9. Each subcategory contains over 500 VQA instances, ensuring a robust representation across all areas, which supports comprehensive model evaluation and facilitates in-depth performance analysis. We provide an overview of the sample sizes and detailed original label information for the seven classification tasks within the BCNB dataset in Table 10.

## A.4 MULTIMODAL DATASET COMPARSION

Recently, several multimodal pathology datasets have been introduced for pathology applications. However, these datasets are often constrained in both scope and scale, as they primarily focus on either patch-level analysis or limited available data. In contrast, our proposed SlideInstruction and

SlideBench, provided as open-source resources, significantly expand the dataset size while enhancing its versatility, as shown in Table 11.

Table 11: Comparisons of our datasets with other pathology datasets.

| Dataset | Level | Data Type | Curation Type | Scope | Number # | Availability |
|---|---|---|---|---|---|---|
| PathChat (Lu et al., 2024b) | Patch | Patch and Q/A pairs | Human+GPT | - | 257,004 | ✗ |
| Quilt-Instruct (Seyfioglu et al., 2024) | Patch | Patch and Q/A pairs | GPT | - | 107,131 | ✓ |
| WSI-VQA (Chen et al., 2024a) | Slide | WSI and Q/A pairs | GPT | - | 8,672 | ✓ |
| PathText (Chen et al., 2023) | Slide | WSI-Caption pairs | GPT | - | 9,009 | ✓ |
| HistGen (Guo et al., 2024) | Slide | WSI-Reports pairs | GPT | - | 7,753 | ✓ |
| Prov-Path (Xu et al., 2024a) | Slide | WSI-Reports pairs | GPT | - | 17,383 | ✗ |
| CR-PathNarratives (Zhang et al., 2023) | Slide | WSIs with multi-modal annotations | Human | - | 174 | ✗ |
| PathAlign (Ahmed et al., 2024) | Slide | WSI-Reports pairs | Human | - | 354,089 | ✗ |
| Our SlideInstruction | Slide | WSI and Q/A pairs | GPT | 13 | 179,935 | ✓ |
| Our SlideBench | Slide | WSI and Q/A pairs | Human+GPT | 13 | 15,835 | ✓ |

# B EXPERIMENT

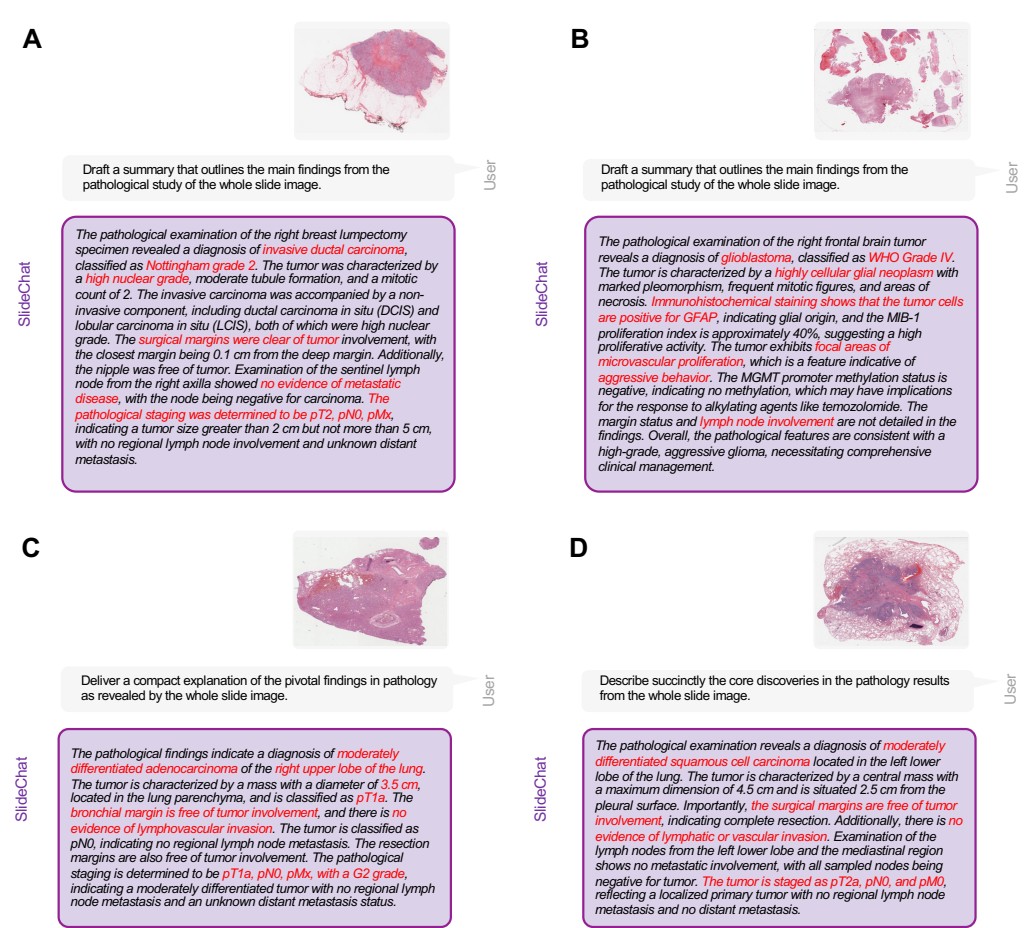

Figure 7: Demonstration of SlideChat's Capability in Whole-Slide Image Captioning.

## B.1 IMPLEMENTATION DETAILS

We preprocessed each WSI by segmenting it into 224 × 224 nonoverlapping patches at a 20× magnification level, excluding background regions. We implemented our model using the Xtuner (Contributors, 2023) toolkit and trained it across two stages on 8 × NVIDIA A100 GPUs. The training

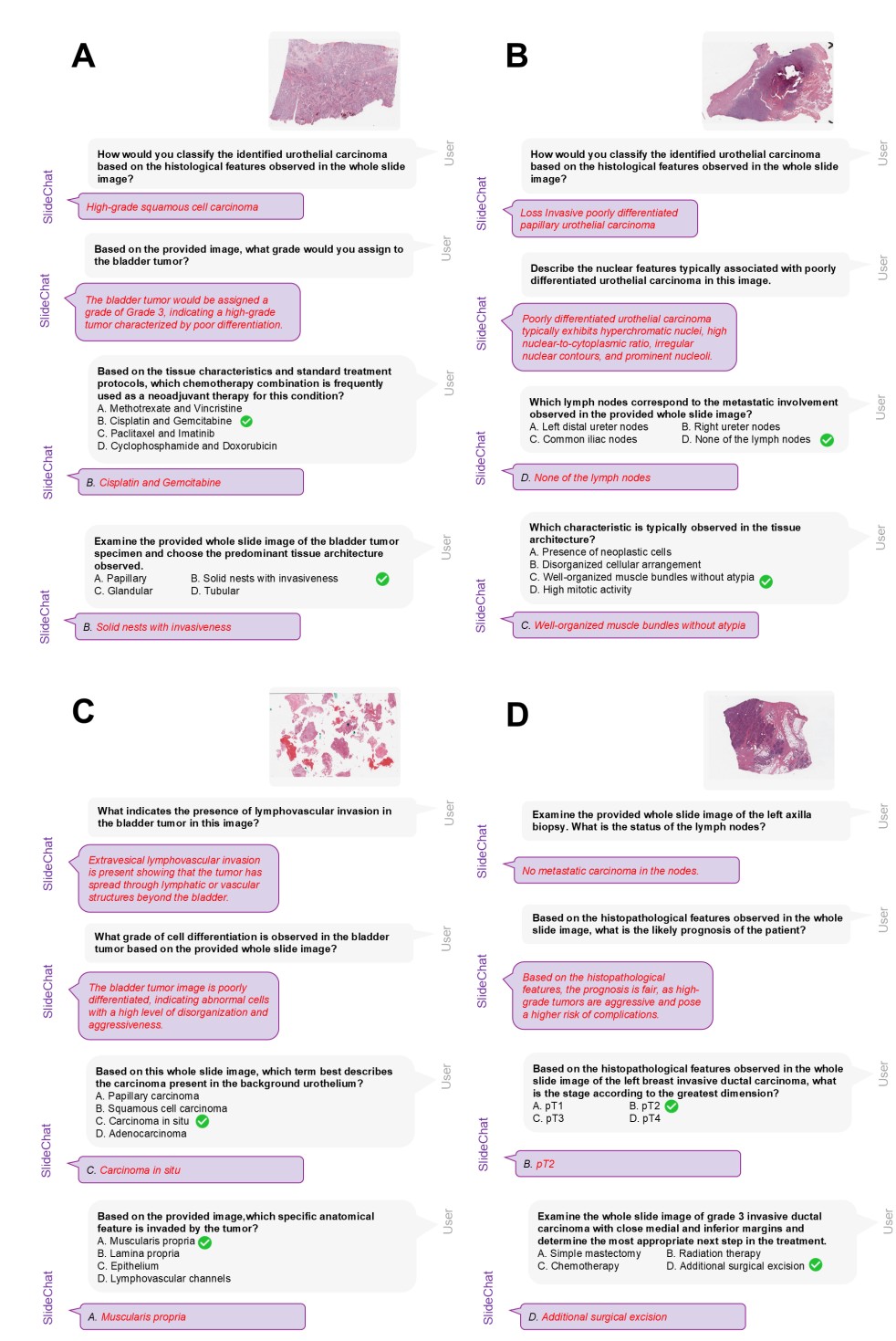

Figure 8: Demonstration of our SlideChat for answering various questions based on the WSI.

process consists of an alignment phase followed by instruction fine-tuning: Stage 1: We freeze the LLM and train the Projection and Slide Encoder with WSI-caption data for 3 epochs, using a learning rate of 0.001. Stage 2: We unfreeze the LLM, Slide Encoder, and Projection, training the model

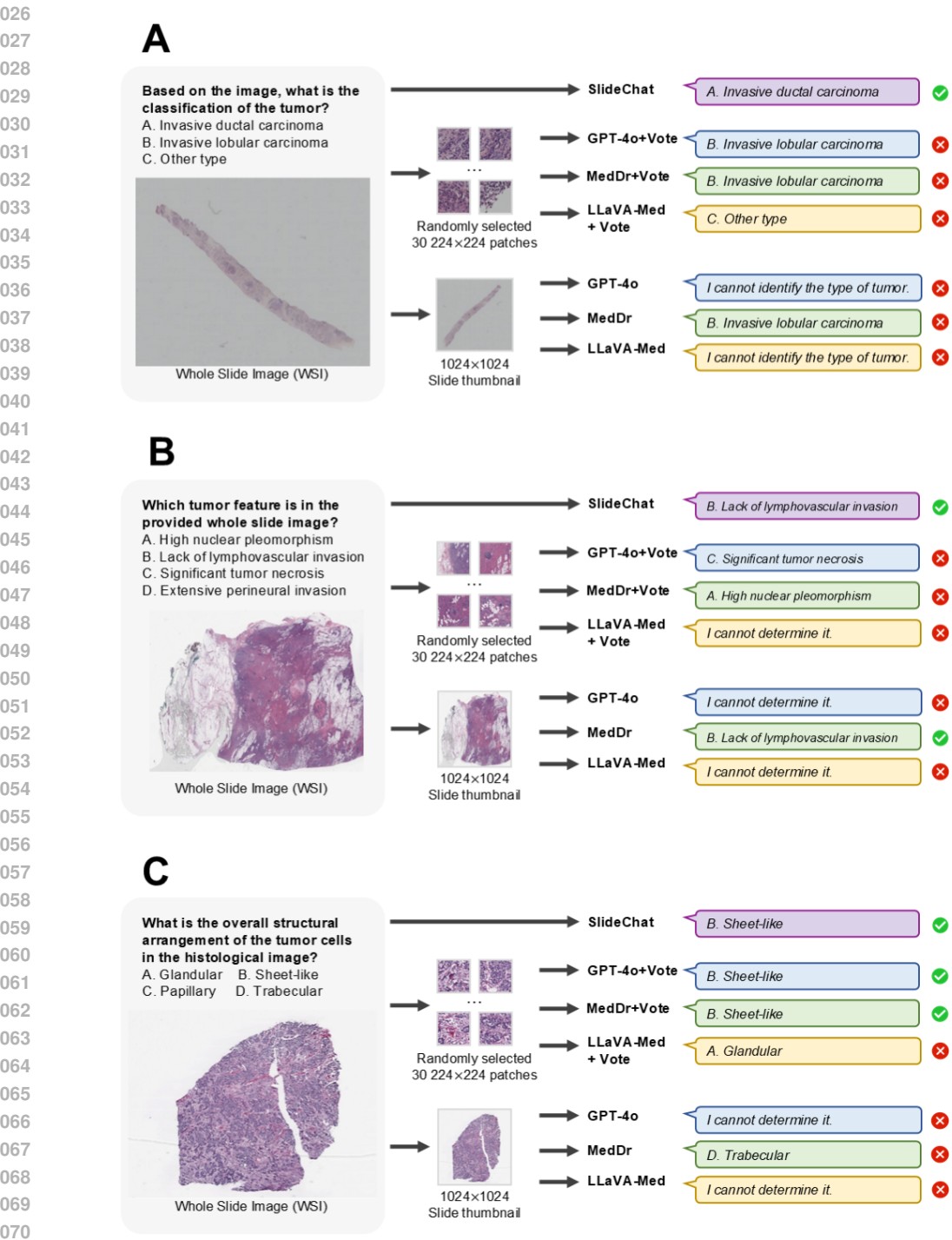

Figure 9: Comparing model outputs on SlideBench.

on WSI instruction-following data for 1 epoch, with a learning rate of 0.00002. Both stages are optimized using AdamW.

## B.2 ABILITY SHOWCASE

### B.2.1 CAPTIONING ABILITY

The examples shown in Figure 7 illustrate the capability of our model, SlideChat, to effectively perform whole-slide image captioning tasks. SlideChat demonstrates its proficiency in generating detailed and contextually accurate summaries for complex pathological whole-slide images, accurately capturing key clinical findings and pathological features. Whether summarizing broad findings, explaining pivotal details, or highlighting core results, SlideChat showcases an advanced understanding of whole-slide images, providing concise yet informative reports that align with clinical terminology and expectations.

### B.2.2 VQA ABILITY

Figure 8 showcases the conversational examples of SlideChat, demonstrating its ability to accurately answer a range of questions based on WSIs, covering diverse aspects such as histological classifications, tumor grading, lymph node involvement, and treatment decisions. SlideChat effectively interprets complex pathological data, engages in nuanced question-and-answer exchanges, and delivers clinically relevant responses. This reflects its potential as an intelligent assistant capable of supporting pathologists in diagnostic decision-making by providing insightful, context-aware dialogue grounded in visual pathology data.

### B.2.3 COMPARING MODEL OUTPUTS

Figure 9 presents a comparative analysis of the outputs from SlideChat and other models within SlideBench. The examples illustrate SlideChat's remarkable capacity to precisely classify tumors, identify distinct histological features, and describe the structural organization of tumor cells from WSIs. SlideChat demonstrates a unique proficiency in capturing both local and global features—seamlessly integrating detailed microscopic characteristics with broader contextual understanding to deliver accurate and clinically meaningful interpretations. In contrast, existing models are limited to processing small pathology images, often yielding ambiguous or incorrect classifications. This underscores SlideChat's advanced capability in comprehending whole-slide images by incorporating both intricate details and a comprehensive visual perspective.

## B.3 DETAILED TEST PERFORMANCE

| Method | Input | SlideBench-VQA(TCGA) **Microscopy** | | | | |
| --- | --- | --- | --- | --- | --- | --- |
| | | Tissue Architecture and Arrangement | Tumor Characteristics | Cytomorphological Characteristics | Histopathological Changes | Overall |
| Random | Text | 23.70 | 22.42 | 23.63 | 27.80 | 24.44 |
| GPT-4 | | 40.83 | 40.28 | 41.71 | 37.46 | 39.62 |
| GPT-4o | Patch | 65.94 | 66.20 | 60.10 | 59.23 | 62.89 |
| MedDr | | 75.04 | 75.78 | 70.10 | 72.23 | 73.30 |
| LLaVA-Med | | 50.04 | 40.63 | 40.38 | 56.95 | 47.34 |
| GPT-4o | Slide (T) | 37.07 | 38.76 | 39.93 | 37.60 | 38.28 |
| MedDr | | 71.58 | 71.27 | 69.87 | 69.05 | 70.48 |
| LLaVA-Med | | 51.80 | 45.02 | 36.27 | 49.01 | 45.82 |
| SlideChat | Slide | 88.07 (+13.03) | 87.01 (+11.23) | 88.02 (+17.92) | 87.36 (+15.13) | 87.64 (+14.34) |

| Method | Input | SlideBench-VQA(TCGA) **Diagnosis** | | | | | |
| --- | --- | --- | --- | --- | --- | --- | --- |
| | | Disease Detection | Disease Classification | Staging | Grading | Differential Diagnosis | Overall |
| Random | Text | 25.82 | 24.06 | 24.14 | 26.12 | 24.40 | 24.91 |
| GPT-4 | | 27.12 | 31.07 | 22.27 | 27.45 | 38.70 | 29.09 |
| GPT-4o | Patch | 50.27 | 55.94 | 39.94 | 39.66 | 49.66 | 46.69 |
| MedDr | | 59.11 | 61.11 | 48.66 | 52.97 | 68.83 | 57.78 |
| LLaVA-Med | | 37.25 | 28.57 | 30.41 | 20.71 | 47.27 | 32.78 |
| GPT-4o | Slide (T) | 22.95 | 26.76 | 18.06 | 21.06 | 27.82 | 23.10 |
| MedDr | | 54.29 | 56.40 | 48.66 | 43.52 | 61.61 | 52.47 |
| LLaVA-Med | | 27.87 | 25.19 | 24.07 | 24.96 | 36.18 | 27.58 |
| SlideChat | Slide | 80.90 (+21.79) | 76.12 (+15.01) | 68.41 (+19.75) | 68.39 (+15.42) | 73.72 (+4.89) | 73.27 (+15.49) |

| Method | Input | SlideBench-VQA(TCGA) **Clinical** | | | | |
|---|---|---|---|---|---|---|
| | | Treatment Guidance | Biomarker Analysis | Risk Factors | Prognostic Assessment | Overall |
| Random | Text | 23.62 | 31.87 | 24.36 | 24.33 | 26.44 |
| GPT-4 | | 49.98 | 44.63 | 46.46 | 39.64 | 45.00 |
| GPT-4o | Patch | 64.18 | 57.99 | 76.99 | 66.64 | 66.77 |
| MedDr | | 74.18 | 82.99 | 82.43 | 60.66 | 74.25 |
| LLaVA-Med | | 62.04 | 53.98 | 53.04 | 26.54 | 47.96 |
| GPT-4o | Slide (T) | 50.00 | 50.08 | 44.16 | 32.64 | 43.42 |
| MedDr | | 71.43 | 84.51 | 78.92 | 60.24 | 72.80 |
| LLaVA-Med | | 50.50 | 48.01 | 48.90 | 19.88 | 40.84 |
| SlideChat | Slide | 83.42 (+9.24) | 89.04 (+4.53) | 91.71 (+9.28) | 74.93 (+8.29) | 84.26 (+10.01) |

| Method | Input | SlideBench-VQA(BCNB) | | | | | | | |
|---|---|---|---|---|---|---|---|---|---|
| | | Tumor Type | ER Type | PR Type | HER2 Type | HER2 Expression | Histological Grading | Molecular Subtype | Overall |
| Random | Text | 23.82 | 24.48 | 25.05 | 25.05 | 24.39 | 24.41 | 23.63 | 24.40 |
| GPT-4 | | 0 | 0 | 0 | 0 | 0 | 0 | 0 | 0 |
| GPT-4o | Patch | 34.69 | 77.50 | 63.51 | 36.95 | 23.95 | 28.63 | 23.15 | 41.43 |
| MedDr | | 45.46 | 23.53 | 25.99 | 71.81 | 22.73 | 30.28 | 15.49 | 33.67 |
| LLaVA-Med | | 23.95 | 36.62 | 40.19 | 50.76 | 23.72 | 18.99 | 15.05 | 30.10 |
| GPT-4o | Slide (T) | 0 | 0 | 0 | 0 | 0 | 0 | 0 | 0 |
| MedDr | | 28.92 | 45.84 | 25.71 | 72.68 | 20.65 | 29.96 | 23.88 | 35.48 |
| LLaVA-Med | | 0.01 | 0.01 | 0.02 | 0.02 | 0 | 0 | 0 | 0.01 |
| SlideChat | Slide | 90.17 (+44.71) | 78.54 (+1.04) | 68.81 (+5.3) | 71.93 (-0.75) | 25.05 (+0.66) | 23.11 (-7.17) | 17.49 (-6.39) | 54.14 (+12.71) |

### B.3.1 PERFORMANCE ON SLIDEBENCH-VQA (TCGA)

The results presented in the tables demonstrate a comprehensive evaluation of SlideChat's performance on SlideBench-VQA (TCGA) in comparison to other existing models across microscopy, diagnosis, and clinical tasks. In microscopy, SlideChat significantly outperforms its counterparts, achieving a notable overall accuracy improvement of 14.34 points over the nearest model. This strong performance is consistent across sub-tasks, such as tissue architecture analysis, tumor characteristics identification, and cytomorphological assessment, showcasing SlideChat's advanced capability to analyze both detailed cellular structures and broader histopathological changes. In the diagnostic tasks, SlideChat also demonstrates superior accuracy, with an overall gain of 15.49 points, excelling in disease detection, classification, staging, grading, and differential diagnosis. The clinical analysis results further validate the model's strength, with SlideChat outperforming other methods by 10.01 points overall, particularly excelling in treatment guidance, biomarker analysis, and risk factor assessment. These results illustrate SlideChat's capability to seamlessly handle complex medical data and deliver reliable insights across multiple clinical and diagnostic domains, indicating its potential as a robust tool for comprehensive pathology analysis.

### B.3.2 PERFORMANCE ON SLIDEBENCH-VQA (BCNB)

The evaluation of SlideChat on SlideBench-VQA (BCNB), a real-world dataset designed for zero-shot testing, further underscores its ability to generalize effectively to unseen data. SlideChat demonstrates an overall accuracy improvement of 12.71 points compared to other models, showcasing its ability to generalize well across diverse and complex breast cancer-related tasks. SlideChat's performance is particularly strong in identifying tumor type, ER status, PR status, and HER2 status, demonstrating a nuanced understanding of critical histopathological features. Nevertheless, in the more complex tasks of HER2 Expression, Histological Grading, and Molecular Subtype classification, SlideChat still exhibits potential for improvement, highlighting specific areas that warrant further refinement to enhance its overall performance.

