# OpenReview forum: "SlideChat: A Large Vision-Language Assistant for Whole-Slide Pathology Image Understanding"
_ICLR.cc/2025/Conference — ICLR 2025 Conference Withdrawn Submission_

### Official Review · Reviewer_9DhE · 2024-10-28

**Soundness:** 2
**Presentation:** 3
**Contribution:** 3
**Rating:** 3
**Confidence:** 4

**Summary:**

This paper proposed SlideChat, the first vision-language assistant capable of multimodal conversation and response to complex instruction. It addresses the challenges in the Multimodal Large Language Model (MLLM) for pathology like, existing MLLMs often focus on patch level; limited publicly available multi-modal datasets. SlideChat is trained on SlideInstruction, a large-scale multi-modal instruction dataset from TCGA. SlideInstruction contains 4181 WSI-caption pairs and 175753 visual question-answer pairs. This paper also proposed evaluating benchmark SlideBench, including a caption and VQA from TCGA, BCNB, and WSI-VQA datasets. Comparing with SOTA MLLM including GPT-4o, LLava-Med, MedDr. SlideChat achieves state-of-the-art performance on most of tasks.

**Strengths:**

This paper is well-motivated and significant. There are no publicly available large-scale VQA tasks for the whole slide image. Since most of the previous benchmarks and models focused on the classification task or small-scale VQA, there is an urgent need for a multimodal benchmark to evaluate WSI understanding.

SlideBench comprises caption and VQA tasks, benchmarking against various SOTA foundation models.

The paper is well-written and easy to follow.

**Weaknesses:**

Lack of quality control for caption and QA dataset. The only quality control is for SlideBench which includes 5 experts to review the QA dataset. There is no discussion like, what were the criteria for resolving disagreements; how many questions did each expert go through; what are their specialties; are they all familiar with these 10 tumor? Since it's the key foundation of this paper. There must be a rigorous human-in-the-loop system to handle the creations.

Over-reliance on ONE language model (GPT4) from ONE company. The entire dataset is generated by GPT4 which may have the same bias and distribution shift. There is a lack of justification for solely using one model and a lack of diversity options on the other model. How do other language models like Claude do if used for generating captions and questions?

The benchmark claims SlideChat's zero-shot performance on VQA tasks. However, the training set and testing set basically have the same or very similar distribution (BCNB). Compared with the other baseline, SlideChat can never be treated as zero-shot. The real zero-shot results are the performances of baselines.

It's known that the TCGA report is very noisy. They have some reports with only one or two lines of description. They could also contain redundant information about radiology and other modalities. There is no quality control system for the SlideInstruction/Caption dataset which may lead to some hallucinations for the language models.

The handling of one report linked to multiple WSIs is not appropriate. The report will indicate which image they referring to in the corresponding descriptions. Even in the case where they don't specify or only provide a general impression. These are not essentially noisy data. And these three WSI image tokens can be grouped to pair with the report.

Evaluation metrics for medical use. should have less penalty for I don't know. It's also not proper to only report accuracy since some diagnosis questions favor metrics like false positive rate, false negative rate, and F1. It will be better to propose a specific way to handle "I don't know" responses in the evaluation.

**Questions:**

Can you clarify if there is a human-in-the-loop system and how it works for expert filtering?

The dataset and GitHub repo are not available. Since the major contribution of the paper is benchmark. what's the author's plan to release, maintain, and host the dataset?

Please consider in-context-learning for the baseline to have a more fair comparison.

Have you considered not using the generated caption but the raw or cleaned version of the original report for training? This will avoid having hallucinations in the caption generation by GPT.

 The scope choice is unclear. since there is a prior work called [TCGA-report](https://github.com/tatonetti-lab/tcga-path-reports) curated over 9K TCGA reports. Is there a specific reason for just choosing part of it?

**Details Of Ethics Concerns:**

The generated contents are related to human subjects. There is not enough safeguards for the data release of this scale dataset.

---

### Official Review · Reviewer_h9zZ · 2024-11-02

**Soundness:** 2
**Presentation:** 3
**Contribution:** 2
**Rating:** 3
**Confidence:** 5

**Summary:**

The authors proposed the first slide-level MLLM for CPATH and curated a slide instruction dataset for training and a slide benchmark for evaluation that includes two main types of tasks, captioning and VQA. Methodology-wise, with patch features from CONCH, LongNet is used to integrate patches into slide-level global features, which are subsequently passed forward to a MLLM, LLaVA, for instruction-tuning.

**Strengths:**

1. This is the first slide-level “LLaVA”-style MLLM unifying both captioning and VQA tasks in CPATH, although it is not the first attempt at the slide-level for each individual task. For example, WSICaption [1] (a.k.a. MI-Gen in this manuscript) is proposed for captioning, and WSI-VQA [2] is proposed for VQA in CPATH.
2. Built upon a public dataset, TCGA, the largest slide-level VQA dataset was curated to train the MLLM.
3. On VQA tasks, the proposed model outperformed general MLLM and some medical MLLM overall, while it cannot surpass MI-Gen in captioning tasks which was trained on around 9K captioning data.

[1] WsiCaption: Multiple Instance Generation of Pathology Reports for Gigapixel Whole-Slide Images, MICCAI, 2024

[2] WSI-VQA: Interpreting Whole Slide Images by Generative Visual Question Answering, ECCV, 2024

**Weaknesses:**

1. Experiments are **NOT** adequate in the following aspects:

    a. To validate the incremental contribution of “slide-level” MLLM, patch-level MLLMs, such as Quilt-LLaVA [3], should be compared. I understand patch-level MLLMs cannot handle the long-sequence input of a slide. However, a global feature (e.g. mean/max pooling of patch features, or the slide-level features extracted by whole-slide pathology foundation models, such as Prov-GigaPath [4] or mSTAR [5]) can be used as the visual feature and fed into patch-level MLLMs.

    b. Performance on task-specific applications should be presented. In clinical practice, accuracy is paramount. Although the authors claimed that pathology foundation models rely on finetuning, their predictions are much more precise than MLLMs from results in appendix B.3, especially on BCNB datasets (which is originally a classification dataset) where **SlideChat is even worse than random guess in grading, subtyping and HER2. Please demonstrate classification performance on these tasks to see how much of the gap is between them.

    c. WSI-VQA, a SOTA slide-level VQA model, should be compared in the experiments.

    d. There is no out-of-domain captioning slide dataset for evaluation, which makes me worry if the model is trying to memorize something because the template is quite similar which will make the metrics caring about text-matching become high while the predicted results are wrong.

2. From the perspective of clinical significance, I can't imagine how pathologists would use this tool, as they wouldn't be asking the model these questions. While the captioning task may have some clinical significance, aiding in report writing, its current performance demonstrated in this paper falls short of supporting this requirement.
3. Methodology-wise, it is a bit incremental by simply combining an existing MIL model and LLaVA.
4. Data-wise, captioning data is much less than others, such as WSICaption. Both are curated from TCGA where over 10K cases of WSI-Report pairs are available. Why are there only 4K cases of captioning data?
5. Train/test split ratios should be given.

[3] Quilt-LLaVA: Visual Instruction Tuning by Extracting Localized Narratives from Open-Source Histopathology Videos, CVPR, 2024

[4] A whole-slide foundation model for digital pathology from real-world data, Nature, 2024

[5] A Multimodal Knowledge-enhanced Whole-slide Pathology Foundation Model, arxiv, 2024

**Questions:**

1. I noticed that prompts for cleaning reports include “The summary should not mention the source being a report and should exclude any specific sizes or measurements.” However, there are still some sizes in Figure 7. I believe the model is unlikely to have the capability to predict sizes from slide images. The same goes for positions like “left breast” or “right breast”.
2. The author used GPT to clean report data. How can the prompt ensure all contents in the cleaned report are observed in the slide visually?

---

### Official Review · Reviewer_3Xjz · 2024-11-02

**Soundness:** 2
**Presentation:** 3
**Contribution:** 2
**Rating:** 3
**Confidence:** 5

**Summary:**

The paper presents a slide-level MLLMs for histopathology utilizing 4181 WSIs from TCGA. The authors use GPT4 to process the patient reports in TCGA to create the question answer pairs for instruction tuning and follow Llava's framework for model training. The model is evaluated on a held-out set from TCGA (for both WSI captioning and VQA) as well as on BCNB (for VQA), and demonstrated better results compared to general / broad medical domain MLLMs including GPT 4o, LLaVA-Med, and MedDr.

**Strengths:**

1. The overall concept and model architecture design is sound.
2. The curation of SlideBench-VQA which is verified by pathologists is valuable.
3. Opensourcing the code and the model checkpoints will be valuable to the community.

**Weaknesses:**

1. Although the paper shows better performance than other baseline MLLMs, none of the baselines are pathology-specific. In WSI-captioning, SlideChat is worse than the only pathology-specific model in the baseline - MI-Gen. It is not clear if SlideChat is better than other pathology MLLMs such as Quilt-Llava in the proposed tasks (including WSI captioning and VQA).
2. The evaluation scheme for patch-level baselines (e.g. GPT 4o) is concerned. While a 1024x1024 thumbnail destroys the aspect ratio of the slide and is too small to make the diagnosis, a random sampling of 30 patches creates randomness between the run and may not well represent the diversity of the patches within the slide, given a slide may have 10^4 224x224 patches (e.g. the diagnostic patch may not be included in the sampled patches). Instead, prototypical-based patch selection methods (which can be as simple as feature-based clustering methods) can be considered for a fair comparison.
3. The small scale of the training (only on 4181 WSIs) limits the diversity of the slides seen during training. How does the model do on out-of-distribution data (such as non-cancer cases or rare diseases e.g. EBRAINS [1]) in terms of diagnosis and IDH status prediction (which can either be in the form of VQA or WSI captioning)?
4. There is no failure case / hallucination analysis, which is especially critical for medical MLLMs. People can gain a better understanding of the model's limitations if the authors can show and analyze when and why model fails, and how does the model hallucinate.
5. There is lack of comparison with supervised baseline on BCNB given the prediction of ER, PR, and HER2 status are standard WSI classification tasks which are commonly included in existing patch-level foundation models [2]. How does the model compare to ABMIL trained on CONCH features?

[1] The digital brain tumour atlas, an open histopathology resource. Scientific Data.
[2] Phikon-v2 A large and public feature extractor for biomarker prediction. arXiv:2409.09173

**Questions:**

1. There are around 10k reports available in TCGA, why do you only use 4181 for training and 734 for testing?
2. What is the distribution of diagnosis / detailed cancer types of the training and testing slides?
3. TCGA can have strong batch effects from submission sites. When you create the split, did you check if your split is site-preserved? If no, the test set (which is then "in-domain") may not well represent the model's generalizibility.
4. For WSI captioning, how does the model compare with other slide-level VLMs such as PRISM [1]?

[1] PRISM: A Multi-Modal Generative Foundation Model for Slide-Level Histopathology. arXiv:2405.10254

---

### Official Review · Reviewer_NBwf · 2024-11-03

**Soundness:** 2
**Presentation:** 3
**Contribution:** 2
**Rating:** 3
**Confidence:** 4

**Summary:**

The authors select a subset of 4,181 pathology whole slide images (WSIs) from TCGA (3,294 patients) and prompt GPT4 to process the patient reports to create WSI-level captions as well as both open-ended / closed-ended 176k (visual) question answer pairs for training and evaluating multimodal large language model specialized for pathology WSIs. After holding a set of WSIs and there associated captions  (SlideBench-Caption) and close-ended VQA data points for testing (SlideBench-VQA (TCGA)), the remaining visual question answer pairs are used for training a model, SlideChat that combines a frozen patch-level feature encoder (CONCH), a slide-encoder based on the LongNet architecture, and a pretrained LLM (e.g. Qwen 2.5-Instruct) for VQA.

SlideChat is compared to general purpose MLLMs like GPT4o as well as medical-specific MLLMs that were not specially trained for pathology or pathology WSIs, and is found to achieve superior performance on the held-out SlideBench-VQA (TCGA) subset as well as an external breast-specific WSI dataset called BCNB (SlideBench-VQA (BCNB)). The captioning performance on SlideBench-Caption was found to be slightly worse but comparable to MI-Gen, a pathology-specific VLM trained for captioning WSIs.

**Strengths:**

The authors main contribution include performing experimental validation of a pathology-slide VQA architecture that combines a pretrained patch-level encoder, a slide-level encoder and a pretrained LLM, as well as releasing the model checkpoint, and associated training and benchmark data (4.2k captions and 176k question answer pairs). On the closed-ended TCGA VQA evaluation set, the authors involved pathologists to review the questions to ensure adequate quality.

**Weaknesses:**

The main weakness of the study lies in the lack of comparison with pathology specific baselines in evaluation. In particular, while authors chose Llava-Med and MedDr as biomedical specialized MLLMs, neither of which are specifically trained for pathology. On the other hand, domain-specific MLLMs such as PA-LLava (https://github.com/ddw2AIGROUP2CQUPT/PA-LLaVA) and Quilt-llava (https://openaccess.thecvf.com/content/CVPR2024/papers/Seyfioglu_Quilt-LLaVA_Visual_Instruction_Tuning_by_Extracting_Localized_Narratives_from_Open-Source_CVPR_2024_paper.pdf) were not evaluated. This may in turn, over-inflate SlideChat's on pathology-specific VQA relative to other methods. The methodology of evaluating general purpose MLLMs like GPT-4o on just a thumbnail or randomly sampled, small crops completely independently before majority vote is also somewhat questionable, and would benefit from further experimentation to make the comparison more convincing.

Additionally, while the release of the 176k VQA question answer pairs is likely still potentially useful for the community (especially the release of SlideBench-VQA, which has been reviewed by expert pathologists to ensure quality), it is difficult to assess its impact relative to other datasets already available in the community, given it stems from just 3,294 patients from TCGA. The previous MI-Gen work, which the authors cite, have already OCRed / used GPT to clean-up ~10k TCGA WSI captions and have made them publicly available - a significantly larger dataset than the 4.2k captions to be released by the authors.

**Questions:**

1. Is the slide-encoder trained from scratch or initialized from the checkpoint released by https://doi.org/10.1038/s41586-024-07441-w?
2. What happens after the slide-encoder? Do you use just the global (i.e. slide-level) token as input to the projection / LLM, i.e. number of tokens = 1 or do you use all contextualized patch tokens? i.e. number of tokens = number of patches?
3. Is there any reason why 224 x 224 was chosen as the patch size for the patch-level encoder when the original CONCH encoder was trained at a patch size of 448 x 448?
4. How does Slidechat's performance compare to publicly available pathology-specific MLLMs like PA-Llava and Quilt-llava referenced in the previous section?
5. In the comparison with GPT4o / other MLLMs, why is 1024 x 1024 chosen as the input size in the "slide thumbnail" experiments but only 30 different 224 x 224 was chosen in the majority voting setup. Why not directly choose multiple large crops (e.g. 1k x 1k) as inputs to GPT4o API to ensure sufficient context is provided given that GPT4o support image dimensions up to 2k and supports a maximum context window of 128k?

---

### Official Review · Reviewer_CGun · 2024-11-04

**Soundness:** 3
**Presentation:** 3
**Contribution:** 3
**Rating:** 5
**Confidence:** 4

**Summary:**

This paper introduces SlideChat, starting with the development of SlideInstruction, which features 4.2K captions and 176K visual question-answer pairs to train a general MLLM for WSIs in computational pathology. SlideChat is capable of interpreting gigapixel WSIs, addressing the limitations of existing multimodal models that often focus on patch-level analysis. Additionally, the authors developed SlideBench, a multimodal benchmark for evaluating SlideChat’s performance across various clinical scenarios, achieving state-of-the-art results on 18 out of 22 tasks. This work holds promise for advancing research and applications in computational pathology, with plans to release the models and datasets as open-source resources.

**Strengths:**

1. The application of MLLMs to WSIs is valuable and the authors conduct extensive experiments to assess the proposed SlideChat.
2. The constructed SlideBench and SlideInstruct are valuable resources for the community.

**Weaknesses:**

1. Generating 175,754 visual question-answer pairs from 4,181 WSI-caption pairs results in nearly 40 QAs per caption. However, the reports for these WSIs often contain only a few key points (around 3-4 important points per report), which can lead to many ineffective or redundant QAs.
2. There may be some overlap between SlideBench-VQA and WSI-VQA, so the authors should clarify this. If any WSIs from the WSI-VQA test set appear in the SlideBench-VQA training set, it could result in an unfair comparison.
3. Comparing other models that use WSI-thumbnails or randomly select 30 patches to represent the WSI is not a fair approach, as both WSI-thumbnails and the limited selection of patches typically lack critical diagnostic information.  I’m also curious about the effects of selecting more patches, say 300, instead of just 30.
4. A comparison with the previous open-source slide-level MLLM, PRISM [1], is necessary.

[1] Shaikovski G, Casson A, Severson K, et al. PRISM: A Multi-Modal Generative Foundation Model for Slide-Level Histopathology[J]. arXiv preprint arXiv:2405.10254, 2024.

**Questions:**

1. Verify the diversity of the approximately 40 generated VQAs for each WSI.
2. Clarify whether any WSIs in the WSI-VQA test set appear in the SlideBench-VQA training set, as this could lead to an unfair comparison.
3. Conduct experimental analysis with PRISM, a previous open-source slide-level MLLM, to benchmark SlideChat's performance.

---

### Note · Authors · 2024-11-13

I have read and agree with the venue's withdrawal policy on behalf of myself and my co-authors.